

# Evaluating Models' Response Of Tropical Low Clouds to SST Forcings Using CALIPSO Observations

Gregory Cesana[1,2,4], Anthony D. Del Genio[2], Andrew S. Ackerman[2], Maxwell Kelley[3,2], Gregory Elsaesser[4,2], Ann M. Fridlind[2], Ye Cheng[1,2] and M.-S. Yao[3,2]

[1] Columbia University, Center for Climate Systems Research, Earth Institute, New York, NY
[2] NASA Goddard Institute for Space Studies, New York, NY.
[3] SciSpace LLC, Institute for Space Studies, New York, NY
[4] Columbia University, Department of Applied Physics and Applied Mathematics, New York, NY

*Correspondence to*: Gregory V. Cesana (Gregory.cesana@columbia.edu)

**Abstract.** Recent studies have shown that in response to a surface warming, the marine tropical low-cloud cover (LCC) as observed by passive sensor satellites substantially decreases, therefore generating a smaller negative value of the top-of-the-atmosphere cloud radiative effect (CRE). Here we study the LCC and CRE interannual changes in response to sea surface temperature (SST) forcings in the GISS Model E2 climate model, a developmental version of the GISS Model E3 climate model, and in 12 other climate models, as a function of their ability to represent the vertical structure of the cloud response to SST change against 10 years of CALIPSO observations. The more realistic models (those that satisfy the observational constraint) capture the observed interannual LCC change quite well ($\Delta$LCC/$\Delta$SST = -3.49 ±1.01 % K$^{-1}$ vs. $\Delta$LCC/$\Delta$SST$_{obs}$ = -3.59 ±0.28 % K$^{-1}$) while the others largely underestimate it ($\Delta$LCC/$\Delta$SST = -1.32 ± 1.28 % K$^{-1}$). Consequently, the more realistic models simulate more positive shortwave feedback ($\Delta$CRE/$\Delta$SST = 2.60 ±1.13 W m$^{-2}$ K$^{-1}$) than the less realistic models ($\Delta$CRE/$\Delta$SST = 0.87 ±2.63 W m$^{-2}$ K$^{-1}$), in better agreement with the observations ($\Delta$CRE/$\Delta$SST$_{obs}$ = 3.05 ±0.28 W m$^{-2}$ K$^{-1}$), although slightly underestimated. The ability of the models to represent moist processes within the planetary boundary layer and produce persistent stratocumulus decks appears crucial to replicating the observed relationship between clouds, radiation and surface temperature. This relationship is different depending on the type of low cloud in the observations. Over stratocumulus regions, cloud top height increases slightly with SST, accompanied by a large decrease of cloud fraction, whereas over trade cumulus regions, cloud fraction decreases everywhere, to a smaller extent.

## 1 Introduction

Low-level clouds are ubiquitous in the tropics. Their presence is tied to a combination of large-scale atmospheric circulation and sea surface temperatures (SSTs), which affect temperature and moisture differences between the surface and the free troposphere (e.g., Bretherton et al., 2013; Klein and Hartmann, 1993). While the underlying processes are not fully understood,



recent observationally-based studies confirm that low-cloud cover (LCC) and SST are negatively correlated (e.g., Mc Coy et al., 2017; Myers and Norris., 2015; Qu et al., 2015).

Therefore, in a warming world, all else being equal, marine boundary layer clouds are expected to dissipate somewhat, which will result in more incoming solar radiation, reinforcing the surface warming through a positive feedback. However, there is no consensus in general circulation models (GCMs) on whether the low-level cloud amount will increase or decrease in future climate projections (Klein and Hall 2015). Moreover, not all models are able to reproduce the observed loss of low-level cloud in response to increased surface temperatures in present day climate, and the majority continue to underestimate the low-level cloud amount (e.g., Cesana and Waliser, 2016; Zhang et al., 2005). Added together, these problems limit our confidence in future climate projections.

As a result, recent efforts have been devoted to evaluating climate models against these observations (e.g.; Klein and Hall 2015; Myers and Norris, 2016; Qu et al., 2015; Mc Coy et al., 2017). This is based on the assumption that models must reproduce the LCC-SST relationship in the current climate as a necessary but not sufficient condition to have confidence in their ability to simulate a more realistic future climate change in regions dominated by low clouds, although there is no guarantee that current climate variability itself is indicative of longer term climate changes (e.g., Marvel et al., 2018). Their results suggest that models that are in better agreement with observations in this way are those with a higher climate sensitivity – i.e., a greater warming of surface temperatures in the future compared to the present-day climate.

All these studies used passive sensor measurements to study this relationship and evaluate the models, because they provide good spatial and temporal coverage along with a long record, which reduces uncertainty in the LCC-SST relationship. However, the space-borne passive instruments typically cannot resolve the vertical extent of clouds and miss some clouds that are shielded by higher clouds. In comparison, the vertical structure of cloud changes in response to surface temperature variations has received far less attention in climate models (i.e., Myers and Norris, 2015). Yet, the 2-dimensional cloud amount as seen from space (i.e., LCC) may hide compensating errors in cloud amount at different levels and does not document the thickness of the cloud. Recent literature has shown the importance of knowing the vertical structure of low clouds to improve our understanding of how clouds may respond to climate change. For example, a deepening of the planetary boundary layer (PBL) – characterized by an increase of the cloud top heights in the low levels – allows more dry air from the free troposphere to be mixed into the PBL, subsequently reducing the cloud amount and therefore generating a positive cloud feedback (Sherwood et al., 2014). It is also hypothesized that the shallowness of the low-cloud layer in the present-day climate may be used as an emergent constraint on GCMs (Brient et al., 2015). Lastly, in addition to other information (e.g., horizontal extent), ascertaining the vertical structure of low clouds could also help discriminating the cumulus clouds from the stratocumulus clouds, the former typically having higher cloud top and variability (e.g., Nuijens et al., 2015). These examples emphasize the need for further evaluation of the vertical structure of clouds in the present day and how it will evolve in a warmer climate.



Thus, active remote sensing instruments can potentially provide important information about the dominant low cloud regimes and their responses to perturbations. In addition to providing detailed information on the vertical structure of clouds, the horizontal resolution of the Cloud-Aerosol Lidar and Infrared Pathfinder Satellite Observations (CALIPSO) lidar is typically finer than that of space-borne passive instruments (90 m footprint vs. a few hundred meters to kilometers), allowing a better

detection of fractional cover of cumulus, which are radiatively dominant in many of the subsiding regions of the tropics. On the other hand, CALIPSO is limited to a 2-dimensional swath and thus produces a much smaller sample of clouds than passive instruments. Thus, active and passive techniques are complementary.

Here we propose to characterize and evaluate the response of tropical low clouds and their radiative impact to SST forcings in

two generations of the GISS ModelE GCM, with a focus on vertical structure, using 10 years of CALIPSO satellite and Clouds and the Earth's Radiant Energy System (CERES) measurements. To put this into a larger context, we also assess this relationship for a large sample of other climate models. Finally, we identify the best performing models, based on how well they match the observed vertical structure relationship between tropical low cloud and SST, and compare the cloud cover response to SSTs of these models against the others.

**2 Data**

**2.1 Observations**

We use the GCM-oriented CALIPSO Cloud Product (CALIPSO-GOCCP) version 2.9 (Cesana et al., 2016) for the LCC and the cloud fraction from 2007 to 2016 over a 2.5˚ grid and for 40 levels with 480 m spacing from 0 to 19.2 km. CALIPSO-GOCCP was developed to facilitate the evaluation of cloud properties in GCMs when combined with a lidar simulator (Chepfer

et al., 2008) that uses the same cloud definitions, and ensures a consistent comparison between observations and simulations. The ratio of the Total Attenuation Backscatter signal (ATB) to the molecular ATB – so-called Scattering Ratio (SR) – is computed every 333 m-along-track near-nadir profile for 480 m height intervals. This quantity is a proxy of the presence of particulate matter in a layer. GOCCP-CALIPSO uses a fixed SR threshold to detect clouds (SR>5), for either daytime or nighttime data, regardless of the vertical level. This threshold allows the detection of thin cirrus cloud in the high-troposphere

(McGill et al., 2007) and prevents most false detections of aerosol layers as being cloudy in the PBL (Chepfer et al., 2013). CALIPSO-GOCCP has been validated against in situ (Cesana et al., 2016) and ground-based observations (Lacour et al., 2017). Caveats for this dataset are discussed in Cesana et al. (2016) and in Cesana and Waliser (2016). The cloud threshold used in CALIPSO-GOCCP allows for the detection of optically thin cirrus (McGill et al., 2007) and hence the majority – if not all – of optically thicker PBL clouds except when masked by overlying high clouds in regimes of weak subsidence (e.g., the trade

wind regions). Additionally, strong attenuation by liquid-topped low clouds may generate an underestimation of the cloud fraction underneath, close to the surface (0 to 960 m, e.g., Cesana et al., 2016), although it does not affect cloud cover. To





avoid daytime noise contamination on the lidar signal, we only use nighttime data, however the results using nighttime and daytime data are similar with a slightly larger amplitude of interannual LCC changes (10 % to 15 % larger).

To derive an uncertainty estimate of the relationship between monthly cloud amount change and SST anomalies over several
years, referred to as interannual change, we use four different datasets for the SST: ERAI, Extended Reconstructed SST version 5 (ERSSTv5, Huang et al., 2017), NOAA Optimum Interpolation (OI) SST version 2 (NOAA-OI SSTv2, Reynolds et al., 2002) and Centennial in situ Observation-Based Estimates SST version 2 (COBE-SST2, Hirahara et al., 2014). The uncertainty related to clouds is due to the cloud threshold and the attenuation of the lidar beam. However, these are reproduced in the model via the use of the lidar simulator and therefore does not necessitate further investigation here. The "actual" observed
relationship may be biased low because of the lidar attenuation and the sensitivity of the dataset to the cloud threshold. While lidar-only products of LCC agree with each other (e.g., Chepfer et al., 2013) some disagreements exist in their cloud profiles due to different definitions of cloudy and fully attenuated pixels in their algorithm (Cesana et al., 2016; Chepfer et al., 2013). Additionally, CloudSat-CALIPSO combined products have been shown to retrieve larger cloud fraction in regimes of weak subsidence but these datasets are only available for a short period of time (Mace and Zhang, 2014) and are therefore unsuited
for this study. For radiative fluxes, we use the monthly CERES Energy Balanced and Filled (EBAF) edition 4 dataset (CERES-EBAF 4.0, Loeb et al., 2018). The large-scale circulation ($\omega_{500}$) is obtained from the monthly ERA-interim reanalysis (Dee et al., 2011). All datasets are averaged over a 2.5° horizontal grid and are used over the same time period as CALIPSO-GOCCP. Using a finer grid (1°) does not impact the results (not shown).

## 2.2 Simulations

In this study, we analyze prescribed-SST (Atmospheric Model Intercomparison Project, AMIP) monthly outputs from two generations of the GISS Model GCM. The first one is the GISS-E2 model that was used for the 5[th] Coupled Model Intercomparison Project (CMIP5) (Schmidt et al., 2014). The second one is a developmental version of the GISS-E3 model – tuned to achieve radiative balance (+0.29 W/m²) using some of the cloud parameters described below – that will be submitted to CMIP6 and will undergo additional changes and tunings by then. E3 and E2 differ in many ways that can potentially affect
low clouds:

(1) Layering in lower troposphere: E2 uses a 40 layer vertical grid, whereas these E3 runs use 62 levels with the greatest refinement in the lower atmosphere: at the surface and at 850 hPa pressure, nominal layer thicknesses for E2 are respectively 20 and 35 hPa, and for the 62 layer grid they are 10 and 20 hPa.

(2) Turbulence: the E2 scheme (Yao and Cheng, 2012), which includes nonlocal transport and does not consider moist processes, has been replaced by the scheme of Bretherton and Park (2009) for E3, which includes moist processes in the computation of turbulent fluxes and uses a novel relaxation approach to parameterize the nonlocal transport of TKE within well-mixed regions; the turbulent transfer coefficients it computes are applied to all prognostic variables separately, with



a water-cloud-only saturation adjustment applied immediately after the transport is treated, using the scheme described below for stratiform cloud macrophysics. The Galperin et al. (1988) scheme that is used by the Bretherton and Park (2009) has been replaced by a second-order scheme with a larger critical Richardson number.

(3) Stratiform cloud macrophysics: while designed differently, both E2 and E3 use a diagnostic determination of cloud fraction as a function of grid-mean moisture and a condition-dependent sub-grid variance expressed as a threshold grid-mean relative humidity (RH) for cloud formation. The Sundqvist-type scheme of E2 (Del Genio et al., 1996), applied identically to water and ice clouds, is replaced for E3 by a scheme that uses a triangular probability density function (PDF) to compute water cloud fraction and cloud water mixing ratio (Smith, 1990). For E3, ice cloud fraction is obtained independently via inversion of that PDF scheme (Wilson and Ballard, 1999), with a different variance than for water. For E3 water clouds, different prescribed values of threshold RH determine the width of the PDF for layers that are within and outside well-mixed regions as determined by the turbulence scheme; this distinction is loosely congruent to $U_a$ and $U_b$ in E2 (Schmidt et al., 2014, section 2.5). In E2, suppression of stratiform cloud under conditions favoring convective cloud is primarily through restriction of the maximum possible areal extent of stratiform cloud to a fraction determined by the depth of convection. In E3 the following check is applied instead: if, above the PBL, a hypothetical saturated parcel is conditionally unstable, stratiform cloud is assumed to be meteorologically inconsistent with the stratification and not allowed to form except at 100% grid-mean RH.

(4) Stratiform cloud microphysics: the Sundqvist-type prognostic cloud water parameterization used in E2 (Del Genio et al., 1996) is replaced in E3 by a two-moment microphysics scheme with prognostic precipitation (Gettelman and Morrison, 2015). For our implementation we use a fixed relative dispersion for the gamma size distribution of water droplets following Geoffroy et al. (2010) and the Meyers et al. (1992) expression for deposition mode heterogeneous ice nuclei, and allow homogeneous aerosol freezing to occur (with a prescribed number concentration) when the RH with respect to ice (grid-mean divided by the fractional threshold RH used to define the width of the PDF used for water cloud fraction) exceeds the threshold of Karcher and Lohmann (2002). Cloud droplet concentrations are prescribed with different values over land and ocean.

(5) Moist convection: as in E2, the cumulus category realized for a given environment is a function of dynamically determined entrainment, which is stronger in E3 as described below. The default entrainment efficiency results in a relatively large rate producing shallow cumulus for typical subtropical conditions; this highly entraining plume may grow deeper under more unstable or moister free-tropospheric conditions. As in E2, a fraction of cloud-base mass flux seeds a second plume with a small entrainment rate conducive to deep convection when conditions are diagnosed to be favorable to mesoscale organization. The E3 version in this study relates the less-entraining fraction to the downdraft mass flux forming cold pools, mirroring Del Genio et al. (2015), whose cold pool parameterization also affects the determination of updraft properties at cloud base. This choice reduces the global frequency and shifts the pattern of less-entraining convection compared to E2, which related it to the large-scale vertical velocity. Reformulations of the numerics in E3, targeting layering independence, eliminated inadvertent but systematic reductions of entrainment rate occurring in E2. Other E2 to



E3 convection changes directly affecting lower-tropospheric conditions include (a) rain evaporation above cloud base, a moistening countered by (b) more efficient venting of the PBL, with the restriction that (c) convection may only originate at the top of a turbulent layer as defined in item (3) above.

(6) Convective cloud microphysics: particle size distributions (PSDs) and size-fall speed relationships used in E2 (Del Genio et al., 2005) have been replaced for E3 with field experiment-based normalized gamma PSDs and fall speeds for ice described by Elsaesser et al. (2017); for liquid, the E2 formulations have been replaced with bimodal (cloud and rain) drop size distributions (DSDs) (each DSD provided by Thompson et al. (2008), with a modified shape parameter from Shipway and Hill (2012) for the rain DSD), while droplet fall speed formulations are now provided by Seifert (2008).

We note that the improved representation of stratocumulus in E3 relative to E2 is principally attributable to the implementation of the moist turbulence scheme, together with critical linkages to stratiform cloud macrophysics and moist convection.

To provide context for the GISS model results, we also analyze AMIP simulations from 12 other CMIP5 models (Table 1). Except for GISS-E3 (2007-2015), we use the last 18 years of AMIP simulations (1991-2008). To ensure a fair evaluation, we compare simulated and observed cloud fields through the use of the lidar simulator (e.g., Cesana and Chepfer, 2012) although the relationships found in this study are very similar (in terms of sign and shape) when original cloud fractions are utilized in GISS-E3. The model outputs are monthly means of the CALIPSO low-level cloud fraction and CALIPSO cloud fraction, so-called cllcalipso and clcalipso, respectively. The simulator package (Bodas-Salcedo et al., 2011) uses profiles of model variables (temperature, pressure, mixing ratios and cloud fraction) in each longitude-latitude grid box for each time step, divides them into sub-columns to account for sub-grid scale variability (Klein and Jakob, 1999) and mimics the lidar simulator signal (Chepfer et al., 2008). Then, the simulated lidar signal is interpolated to the CALIPSO-GOCCP vertical resolution, 40 levels of 480 m thickness between 0 and 19.2 km, and the different diagnostics are computed and accumulated into statistics. A sub-pixel is diagnosed as cloudy when its SR is larger than 5 and low-level clouds are diagnosed in the column whenever a cloudy pixel is present below 3.36 km.

## 3 Method

### 3.1 Definition of low cloud regions

In this work, we focus on the low-level clouds that form over the tropical oceans (between 35˚S and 35˚N) in subsidence regimes defined as having a large-scale pressure vertical velocity at 500 hPa ($\omega_{500}$) greater than 10 hPa/day. This filtering captures most of the stratocumulus and stratocumulus-to-shallow-cumulus transition regions, which are located climatologically within the blue contours in Fig. 1. In the literature, some studies use a 0 hPa/d $\omega_{500}$ threshold (e.g., Myers and Norris, 2015, 2016). Here we choose a more conservative $\omega_{500}$ threshold to minimize areas where high clouds are common and that may mask the detection of underlying low-clouds in the observations. We confirm this by looking at the height at



which the lidar signal becomes completely attenuated, so-called z_opaque (Guzman et al., 2017). The 10 hPa/d threshold almost perfectly encompasses areas where z_opaque is smaller than 2 km (see Fig. S1), meaning that the lidar is able to detect virtually all low clouds in these regions (clouds with cloud top lower than ~ 3 km).

### 3.2 Cloud-SST relationship and observational constraint

Two main goals of our study are to investigate the interannual variation of the vertical cloud fraction (CF) and LCC in response to a change in SST in both the observations and the models, and to use the observed relationship to evaluate the models. By interannual variation we mean the monthly variations over multiple years, a decade in this case. Capturing the mechanisms that govern the change of clouds in response to a surface warming is an essential condition – although not the only one – to predict future climate. Thus, we select the GCMs that produce the most realistic change in cloud profile per K of SST warming.

We refer to these as "constrained" models, in the sense that they are distinguished from other models in our analysis using an observational constraint; we emphasize though that the models have not been changed in response to the observations. We compare the cloud fraction and shortwave (SW), longwave (LW) and net cloud radiation effect (CRE) changes of these models to the others, which we refer to as "unconstrained" models.

To calculate the interannual relationship between SST and cloud amount, we compute the monthly mean of CF and LCC and monthly anomalies of SST after having filtered out all grid boxes where $\omega_{500}$ is lower than 10 hPa/d, referred to as $CF_{sub}$, $LCC_{sub}$ and $SST_{sub,anom}$. Those can be seen as dynamically-based means and anomalies, as opposed to spatially-based anomaly/mean studies that focus on particular regions (e.g., McCoy et al., 2017, Qu et al., 2015). Hence, the cloud response is dominated by the local component rather than the large-scale component (dynamics). It is therefore complementary to

imposing a uniform +4K increase (e.g., Cesana et al., 2017) or an abrupt 4 times $CO_2$ increase (e.g., Brient et al., 2016) that are also significantly affected by dynamical changes. We then linearly regress $CF_{sub}$ and $LCC_{sub}$ against $SST_{sub,anom}$ to obtain the change (Δ) in cloud fraction and low cloud cover per K of SST warming ΔC/ΔSST, where C is either the CF or LCC. Using a centered finite-differencing scheme as in Myers and Norris (2015) instead of a linear regression does not impact the results (not shown).

### 3.3 Assumptions and caveats

By using this method, we make some assumptions that generate some caveats. For example, we assume that the relationship between SST and low cloud amount is time-scale invariant, i.e., the same regardless of the time-scale over which anomalies are calculated. This assumption seems to be supported by several previous studies (e.g., Klein et al., 2017; Mc Coy et al., 2017), but we note that any such relevance to cloud feedback in the regions we study does not necessarily have broader

implications for the global equilibrium climate sensitivity (Caldwell et al., 2018). Moreover, we analyze the effect of SST on clouds by assuming that the cloud effect on the SST is negligible on a monthly time-scale based on previous studies (e.g., deSzoeke et al., 2016; Klein et al., 2017; McCoy et al., 2017). The relatively short period of the time record is another caveat



here. However, the standard deviation (STD) computed using the four SST datasets (or the 5-95 % confidence intervals when using a single SST dataset, not shown) is far smaller than the multimodel mean STD and bias, as shown in section 4. In addition, using a smaller period of time does not change the sign and shape of the results but may change its magnitude (not shown).

Other environmental factors may cause low cloud changes such as the estimated inversion strength or $\omega_{500}$ (Qu et al., 2015; Myers and Norris, 2016). When these factors are held constant the variation of the cloud amount as a function of the SST becomes a partial derivative. Past studies have shown that computing the partial derivative may decrease the magnitude of $\Delta$LCC (e.g., Myers and Norris, 2015; Qu et al., 2015). We find a similar decrease in our study using four of the five

observational datasets of section 4.3 ($\Delta$LCC ~ 20 % smaller, see section 4.3).

As stated earlier, our $\omega_{500}$ filter targets stratocumulus and stratocumulus-to-shallow-cumulus transition regions. Such a definition of low clouds – while extensively used in the literature – does not permit us to distinguish between the two most common low-cloud types, that is to say trade cumulus and stratocumulus, and it also excludes parts of the trade cumulus

regimes that have been argued to be important to overall cloud feedback (weak convective regimes, e.g., Nuijens et al., 2015). As a consequence, our results do not target a specific type of cloud but rather represent the regional-only averaged effect of all types of low clouds. Nevertheless, we attempt to provide some information on the observed interannual changes of low clouds in trade-cumulus and stratocumulus regimes in section 4.3.

## 4 Results

### 4.1 Constraining the vertical response of low-level cloud fraction

Figure 2a shows averaged cloud fraction profiles over the tropical oceans (35˚S to 35˚N) in subsidence regimes ($\omega_{500}$ > 10 hPa/d). In the low levels (z < 3.36 km), both GISS models underestimate the CF. Although GISS-E2's peak (purple line with stars) is slightly larger than E3's (blue line with stars), the shape of the GISS-E3 profile is in better agreement with the observations (two large values at 1.2 km and 1.68 km). In addition, GISS-E3's CF values are in very good agreement with the

observations at 2.16 km and above while they are overestimated in GISS-E2, suggesting an excess of trade cumulus type of clouds. Most of the other models (9/12) also underestimate the CF, yielding a multi-model mean peak ~43% smaller than observed (triangle green line, 11.2 %, vs. circled orange line, 19.6 %, Fig. 2a). In addition, the model behavior is relatively diverse, which highlights the large uncertainty around the simulation of low clouds. The observed shape of the cloud fraction profile – a single peak around 1.2 km – is not captured by all models. Some simulate a double-peak shape, which is likely the

result of the distinct contribution of stratocumulus and trade cumulus clouds, the latter having typically smaller CF and higher cloud top (typically treated by separate parameterizations in a model). Other models show a single peak as in the observations but with a far smaller CF. This could be explained by several reasons: that is a too shallow PBL, a general lack of low clouds





for a given thermodynamic state, a strong masking effect by overlying high clouds or by a larger influence of a convection parameterization over that of the large-scale cloud and turbulence parameterizations that determine stratocumulus clouds.

In Figure 2b, we show the interannual change in CF per K of SST warming (ΔCF/ΔSST) based on a linear regression method

between SST anomalies and CF, as described in Section 3.2. As for the mean cloud profiles, the model responses are quite diverse, generating a very large variability compared to the observed STD. A group of models predict a very small change, which can be either an increase, a decrease or both at different heights. Others models simulate a large increase of CF at cloud top and a large decrease below, i.e., an upward shift rather than a cloud cover change. Finally, the remaining models reproduce the shape of observed change pretty well, that is to say a large decrease below 2 km.

In this study, we assume that i) the physical mechanisms that control the subtropical low-cloud response to warmer surface temperature remain identical across all time scales and ii) those mechanisms are essential to predict the correct subtropical low-cloud change in the future, although they may not necessarily be the only ones (e.g., current climate variability does not include the radiative effect of increased $CO_2$ on cloud-top turbulence). Additional phenomena, e.g., large-scale dynamical

feedbacks that differ on interannual and centennial time scales, could also mitigate or amplify the change. However, we believe that the present-day interannual change in the cloud fraction (ΔCF/ΔSST) is one important test that a model must pass to have confidence in its prediction of future climate. We therefore isolate the change of the low-cloud cover associated with a surface warming as well as the related top-of-atmosphere radiative impact for the subset of models that best reproduce the observed cloud fraction change – i.e., a large CF decrease (< -1 % K$^{-1}$) and no significant CF cloud top increase (< +0.5 % K$^{-1}$) (see Fig.

S2 for details). In the remainder of the manuscript, we will call this category the "constrained models" (6/14, marked with a star in Table 1), represented in blue, and the other models the "unconstrained models" (8/14), represented in purple. The two GISS models fall into each category: the unconstrained category for GISS-E2 and the constrained category for the newest version, GISS-E3.

Overall, the constrained models simulate a larger cloud amount at low levels, in better agreement with CALIPSO, than the unconstrained models (Fig. 2c). In addition to underestimating the low-level cloud amount and its decrease with surface warming, some unconstrained models predict low-level cloud top rising, either because of a deepening of the PBL or due to an increase of the upper cloud fraction peak (Fig. 2d). This cloud-top rising may imply an excess of trade cumuli in the present-day climate in the models having a dual-peak cloud fraction in the low levels (e.g., CCSM4-CAM4, MIROC, MRI, GISS-E2

and MPI, Fig. S2): one large peak close to the surface (stratocumulus type) and another smaller peak above (trade cumulus type).



## 4.2 Consequences for low-cloud cover

In the remainder of the manuscript, we use star shapes in our plots to distinguish the GISS models from the other models and emphasize the effect of cloud parameterization changes with respect to interannual LCC and cloud radiative effect (CRE) changes in a GCM.

Based on this observational constraint, we now investigate how well the models simulate LCC in present-day climate and with a surface warming. Figure 1 shows the LCC maps for the observations and for the two model categories as well as their biases. Although the LCC global means of GISS models are almost identical ($LCC_{E2}$ = 28.5 % and $LCC_{E3}$ = 28.6 %), their spatial patterns (Fig. 1b-d) are completely different (E2 failing to produce any stratocumulus clouds), which results in a very poor

correlation factor for E2 (r = 0.11, the smallest of all 14 models) as opposed to a very good one for E3 (r=0.86, the largest of all 14 models). The reader should also bear in mind that E3 cloud fraction and cloud cover are slightly underestimated in the present study because the simulator is run offline (at daily frequency), which generates lower cloud fractions and cloud covers than the inline version (not shown). The constrained models (Fig. 1h) simulate larger LCC global (and tropical) means (LCC = 30.5 %, r = 0.92), closer to the observations (LCC = 37 %), and also better reproduce the observed LCC pattern than the

unconstrained models (Fig. 1j, LCC = 25.7 %, r = 0.86) and the multimodel mean (Fig. 1f, LCC = 27.8 %, r = 0.90).

We apply the same method as in Section 3.2 to calculate the interannual change in LCC per K of surface warming (Figure 3a and Table 2 first column, ΔLCC/ΔSST. Consistent with the cloud fraction profiles, GISS-E3, the only model being within the observation uncertainty, predicts a decrease of the LCC in response to a local 1K surface warming (-3.55 % $K^{-1}$), like most

models (12/14), as opposed to a small increase for GISS-E2 (0.22 % $K^{-1}$). As the diffence between GISS-E2 and E3, the multimodel spread is significantly large (5.4 % $K^{-1}$, Table 2), which is about two and half times greater than the absolute value of the multimodel mean (-2.25 % $K^{-1}$, Table 2). However, the constrained models simulate a ΔLCC/ΔSST slightly smaller than the observation but within the observational uncertainty (-3.59 % $K^{-1}$ +/- 0.28 % $K^{-1}$) and with a much-reduced spread (-3.49 % $K^{-1}$ +/- 1.01 % $K^{-1}$). The observed ΔLCC/ΔSST is significant as its amplitude is more than three times larger than the LCC

annual standard deviation in the same dynamical regimes (1 % $K^{-1}$).

It is plausible to think that ΔLCC could depend on the initial amount of LCC in a model (e.g., Brient and Bony, 2012). While the difference between GISS-E2 and GISS-E3 is not substantial, comparing this relationship for multiple versions of the GISS-E3 model (run along the course of its development) supports a relationship between ΔLCC and the present-day LCC in

subsidence regions (Fig. 3b). This relationship holds regardless of whether the simulator is used or not. Except for MIROC5, which simulates a present-day LCC almost as large as the observations, the constrained models simulate a larger present-day LCC in subsidence regions (consistent with what was found in Fig. 2). When MIROC5 is set aside, the correlation between the LCC and ΔLCC in Fig. 3 becomes more obvious (r = -0.57 vs. r = -0.40 for all models). One should note that the present-



day LCC could be biased low in some models, due to a too strong shielding effect by overlying high-clouds compared to the observations, possibly affecting the relationship between the present-day LCC and ΔLCC. In the GISS-E3 model, the simulator does not affect ΔLCC (Fig. 3; compare red and black versions of the same symbols), despite its significant impact on the present-day LCC as hypothesized before. In addition, the relationship may be different depending on the type of clouds, since

Fig. 3 does not separate trade cumulus from stratocumulus.

### 4.3 Consequences for annual low-cloud feedbacks

In this section, we further examine the impact of cloud changes on the radiative budget for the same stratocumulus and stratocumulus-to-shallow-cumulus transition regions (over the tropical oceans and based on $\omega_{500}$), using CRE, defined as the difference between the all-sky flux minus the clear-sky flux at the TOA. Figure 4 shows the change in the SW, LW, and net

CRE per K of surface warming referred to as ΔCRE/ΔSST (i.e., dCRE/dSST). A positive ΔCRE/ΔSST implies a warming of the climate system due to clouds when the SST increases; conversely, a negative ΔCRE/ΔSST implies a cooling effect. This quantity may be used as a proxy to characterize cloud feedbacks at the top of the atmosphere (TOA; e.g., Medeiros et al., 2015, Cesana et al., 2017). All observed $\Delta CRE_{SW}/\Delta SST$, $\Delta CRE_{LW}/\Delta SST$ and $\Delta CRE_{NET}/\Delta SST$ are positive, a feature particularly well-captured by GISS-E3, which is surprisingly good for both the SW and LW components of the interannual feedback, while

GISS-E2 gets the sign of the SW component wrong. Both constrained and unconstrained multimodel means (colored triangles) get the correct sign of all three feedbacks although the sign and the magnitude of $\Delta CRE_{NET}/\Delta SST$ vary significantly among the models, mostly driven by the SW component, in agreement with previous studies (e.g., Medeiros et al., 2015, Cesana et al., 2017). Overall, the constrained models perform better than the unconstrained models for all three components, in terms of absolute value and variability. In particular, the unconstrained models largely underestimate the $\Delta CRE_{SW}/\Delta SST$ (0.73 W m$^{-2}$

K$^{-1}$, Table 2 second column), compared to the observations (3.05 +/- 0.28 W m$^{-2}$ K$^{-1}$) whereas the constrained models almost fall within the observed uncertainty (2.60 W m$^{-2}$ K$^{-1}$).

Because of the optical properties of their spherical droplets, low-lying warm marine cloud reflect more sunlight than the underlying ocean surface. As a result, any change in LCC should affect the $CRE_{SW}$ at TOA and one should expect a good

correlation between the two quantities, which is demonstrated in Fig. 4a, with a linear correlation coefficient of -0.94 (excluding the outlier of the calculation). There is little correlation for the LW component whereas for the net component, the correlation is also very large (r = -0.94), driven by the shortwave radiation, confirming its crucial role in determining the cloud feedback spread of CMIP models (e.g., Andrews et al., 2012). Once again, both the magnitude and the variability of the three components is better reproduced by the constrained category of models.

In addition, we analyzed the sensitivity of $\Delta CRE_{SW}$ to ΔLCC by simply computing the ratio between the two quantities as in Klein et al., 2017 (Table 2, third column). GISS-E2 largely overestimates the magnitude of this ratio (by a factor of 10) as do two other models (IPSL-5A and CNRM), that poorly represent the climatological stratocumulus decks. On the other hand,



GISS-E3 stands out among the best models and replicates the observed ratio. Like GISS-E2, the unconstrained models largely overestimate the radiative impact of an LCC loss (-3.13 W m$^{-2}$ %$^{-1}$ compared to the observations (-0.85 W m$^{-2}$ %$^{-1}$) while the constrained models reproduced the observed relationship quite well (-0.74 W m$^{-2}$ %$^{-1}$). The inability of the unconstrained models to simulate a sufficient amount of LCC in the present-day climate may generate a lack of outgoing SW radiation at

TOA, which is compensated by artificially increasing the reflectivity of the clouds during the tuning process in some modeling centers (e.g., Nam et al., 2012).

The constrained models all generate large stratocumulus decks along with a substantial amount of tropical low clouds in non-stratocumulus regions, which seems key to simulating the correct global response of low clouds to surface warming. This

behavior is likely due to the fact that they simulate moist processes in the PBL by either turbulence (e.g., GISS-E3, CESM1-CAM5, GFDL AM3, hadGEM2A, CanAM4), convection (IPSL5B) or both parameterizations (hadGEM2A), in addition to having a stratocumulus decks. This becomes more evident when looking at the evolution of individual models. For example, implementing a more physically-based "moist" turbulence parametrization (following Bretherton and Park, 2009) in the GISS-E3 model changes the sign of $\Delta$LCC/$\Delta$SST and $\Delta$CRE$_{SW}$/$\Delta$SST and brings the model results within the range of uncertainty of

the observations. Similarly, the changes in the IPSL model from version 5A to 5B significantly improved its simulation of the $\Delta$LCC and $\Delta$CRE$_{SW}$ quantities most likely because its "dry" PBL was effectively turned into a "moist" PBL through the implementation of moist shallow convection within the PBL (Rio and Hourdin, 2008), which improved their wind profiles and PBL height (Hourdin et al., 2013), combined with a revision of their turbulence scheme, which improved their representation of stratocumulus clouds. However, the MPI "moist-PBL" model does not fall into the constrained category. Even though its

results are quite close to the observations, the clear overestimation of the cloud frequency above 2.16 km (Fig. S2, likely trade cumulus clouds) alters its $\Delta$CF and leads to a sensitivity of $\Delta$CRE$_{sw}$ to $\Delta$LCC that is too strong. Conversely, the BCC "dry-PBL" model captures $\Delta$LCC and $\Delta$CRE$_{sw}$ variations pretty well (within the range of the constrained models) although its $\Delta$CF is unrealistic. Therefore, the capacity of the models to replicate the observed response of low-level clouds and radiation to warmer surface temperature seems to be tied to whether or not i) they simulate moist processes in the PBL and ii) their

turbulence scheme sustains stratocumulus clouds. Such results also demonstrate that a simple 2D description of the cloud properties – i.e., as seen from space-borne passive sensors – is not sufficient to fully understand and predict how cloud may react to surface temperature forcings and further requires information on the vertical structure of clouds.

**4.4 Discriminating trade cumulus from stratocumulus clouds**

Given the different factors controlling cumulus and stratocumulus clouds, one could expect a different response of each type

of cloud to a surface temperature perturbation. This is further supported by the diverse behavior of modeled $\Delta$LCC/$\Delta$SST, which is correlated with the ability of the models to produce a large amount of stratocumulus or not in the present climate. To verify this, we determine the $\Delta$LCC/$\Delta$SST of trade-cumulus ($\Delta$LCC$_{TrCu}$/$\Delta$SST) and stratocumulus-dominated regions ($\Delta$LCC$_{Sc}$/$\Delta$SST) and their associated $\Delta$CRE$_{SW}$/$\Delta$SST.



Distinguishing cumulus from stratocumulus clouds is particularly challenging in the observations. Climatologically the two cloud types can be separated using k-means clustering of optical thickness-cloud top pressure histograms over GCM grid-sized areas (Chen and Del Genio, 2009), although instantaneous errors can arise, e.g., from overlying clouds. In the PBL, as the inversion strength increases, the moisture tends to increase, leading to larger cloud fractions (e.g., Klein and Hartmann, 1993). This phenomenon explains why lower tropospheric stability (LTS), defined as the difference between potential temperature at 700 mb and the surface, is well correlated with LCC in the observations, over the tropical oceans (e.g., Klein and Hartmann, 1993; Wood and Bretherton, 2006). We verified this relationship using LTS derived from the ERAI reanalysis and CALIPSO-GOCCP LCC. The correlation between the two quantities is 0.65 but decreases when limited to larger LTS values (0.51 for LTS > 15 K, 0.38 for LTS > 17 K). We tried to use LTS-based thresholds to separate stratocumulus decks from other low-level clouds but the method does not work well for monthly climatology (not shown). Besides, only a few models have high LTS-LCC correlations (4/14 larger than 0.6), and for those, the larger LTS do not match the stratocumulus areas. Note that using convective and stratiform cloud fraction (often separated in GCMs) would solve this problem on the model side but such partitioning is not provided in the CMIP5 archive.

Instead, we focus on eight specific regions (Fig. S3) that have been identified as being dominated by either stratocumulus clouds (Sc) or trade cumulus clouds (Cu) in previous studies (e.g., McCoy et al., 2017). This method does not allow a model evaluation as the models may not be able to simulate the correct type of clouds in these regions, regardless of their ability to reproduce the response of each type of cloud to SST variability. Therefore we focus our analysis on observations only. In the literature, all studies referenced before but (Brient and Schneider, 2016) exclusively used passive sensor to derive ΔLCC/ΔSST composites. In contrast, we use CALIPSO-GOCCP, which has a shorter time record and poorer sampling but a greater sensitivity to trade cumulus clouds due to both its narrower horizontal footprint and its better instrument sensitivity to liquid cloud particles. However, because we are using different methods and regions in our study, we included two ISCCP (Qu et al., 2015 and Pincus et al., 2012) and two MODIS (Pincus et al., 2012) estimates of ΔLCC/ΔSST for comparison. The original ISCCP dataset is the same used by Pincus et al. (2012), a GCM-oriented ISCCP dataset prepared to facilitate the evaluation of GCMs (consistent with the ISCCP simulator) based on a subset of ISCCP variables (Rossow and Schiffer, 1999). The second ISCCP dataset, so-called ISCCP Q15, is derived from the first one. A correction for possible masking effect of overlying clouds is applied as in Qu et al. (2015): LCC' = LCC+M / (1-H), where LCC, M and H are the original low-cloud cover, the mid-cloud cover and the high-cloud cover, respectively. Those can be interpreted as low and high uncertainty estimates. Using GISS-E3 and the ISCCP simulator, we found that the corrected LCC' out of the ISCCP simulator is slightly overestimated compared to the original model LCC although the two quantities are highly correlated (see Fig. S5). We excluded data before 1999 (April 1999 to March 2008, 10 years as for CALIPSO-GOCCP) due to artifacts in the dataset. Similarly, the MODIS datasets combine observations from the MODIS Terra and Aqua platforms for model evaluation (also consistent with the MODIS simulator). The daily collection 5.1 files are monthly averaged and a subset of relevant variables are saved. The first





dataset includes the cloud fraction from cloud retrievals, which are the cloud fraction used to derive MODIS cloud properties in the collection 5.1 product. These are pixels that are entirely filled with clouds. On the other hand, the second dataset contains the cloud fraction from the so-called MODIS mask, which includes partially cloud-filled pixels (Pincus et al., 2012; Platnick et al., 2003). Here we used 15 years of data from 2001 to 2016.

Figure 5 shows the $\Delta LCC/\Delta SSTs$ for all datasets using the same cloud regimes based on $\omega_{500}$, and the same four trade cumulus- and four stratocumulus-dominated regions (Fig. S3). Consistent with previous studies (e.g., McCoy et al., 2017; Myers and Norris, 2017; Qu et al., 2015), all datasets agree on a decrease of the LCC for increasing SSTs. The decrease still occurs but to a smaller extent (~ 20 % smaller) when the Estimated Inversion Strength (EIS, Wood and Bretherton, 2006), another supposed low-cloud controlling factor, is held constant (cf., Klein et al., 2017). The overall magnitude of the change is larger

in CALIPSO (-3.59 % K$^{-1}$) than in the passive sensor datasets (-1 to 2.95 % K$^{-1}$, Table 3). When only the stratocumulus regions are considered, all datasets show a larger decrease of the LCC than for the trade cumulus clouds or all tropical low clouds. This may suggest that the overall behavior of clouds is controlled by the Cu (i.e., Bony and Dufresne, 2005), which supposedly cover a larger area of the tropics, although it does not guarantee it as we do not know for sure what type of clouds cover what part of the tropics from the observations. The difference between ISCCP and GOCCP is also relatively smaller in Sc regions

($\Delta LCC_{GOCCP,Sc}/\Delta SST$ = -5.32 % K$^{-1}$ vs. $\Delta LCC_{ISCCP,Sc}/\Delta SST$ = -5.22 and -6.06 % K$^{-1}$ for the two ISSCP products described above) than in the Cu regions ($\Delta LCC_{GOCCP,Cu}/\Delta SST$ = -3.62 % K$^{-1}$ vs. $\Delta LCC_{ISCCP,Cu}/\Delta SST$ = -2.31 and -1.4 % K$^{-1}$, Table 3). Without the east coast of Peru region, MODIS observations also agree well with ISCCP and GOCCP $\Delta LCC_{Sc}$ (within 15%, not shown) although the MODIS mask cloud cover remains smaller for the most part than all other datasets regardless of the

regions selected.

Finally, it is also worth mentioning that the sensitivity of CRE at TOA per unit change in cloud fraction is significantly smaller in magnitude for trade cumulus clouds than for stratocumulus clouds in the three satellite estimates, including CALIPSO-GOCCP (-0.44 W m$^{-2}$ %$^{-1}$ vs. -1.34 W m$^{-2}$ %$^{-1}$), consistent with the fact that trade cumulus clouds are less reflective than

stratocumulus clouds (Table 3). Even though differences in TOA CRE would emerge if one could use CERES-like observations at the CALIPSO horizontal resolution, these biases would remain small (e.g., Ham et al., 2015). In addition, we document the cloud opacity in the two regions using the ratio of opaque cloud cover (fully attenuating the lidar, Guzman et al., 2017) to the total cloud cover, $R_{opacity}$. We find that the stratocumulus $R_{opacity}$ (75.9 %) is 50 % larger than that of trade cumulus regions (50.6 %), confirming the larger optical thickness of clouds in the stratocumulus regions than in the trade

cumulus regions. Overall, all passive sensor estimates of this quantity are larger in magnitude than that of CALIPSO-GOCCP (-0.85 W m$^{-2}$ %$^{-1}$ vs. -1.11 to -3.26 W m$^{-2}$ %$^{-1}$, Table 3), even more so in the trade cumulus regions.

Finally, the vertical response of the CF to a surface warming ($\Delta CF_{all}$, $\Delta CF_{Sc}$, $\Delta CF_{Cu}$) is shown in Figure 6b. In the observations, the low cloud top is lifted up coincident with a large decrease in cloud fraction below in stratocumulus regions (purple line)



while in the trade cumulus regions, the cloud top does not change and the decrease is significantly smaller (green line). Note that because our Sc and Cu are defined by regions rather than actual cloud types, we cannot distinguish between an actual rising of Sc cloud tops and a transition from lower-topped Sc to higher-topped Cu; both may contribute to the behavior of the $\Delta CF_{Sc}$ profile. Depending on low-level cloud top height and the type of cloud, the effect of a surface warming is therefore
different, generating a small decrease for "higher" low-level clouds (Cu) as compared to a larger decrease of the "lower" low-level clouds (Sc) along with an increase of their cloud top height. Thus, favoring one cloud type over the other in the models may result in either an overestimate (too many Sc) or an underestimate of the $\Delta LCC/\Delta SST$ (too many Cu). Because the overall $\Delta LCC/\Delta SST$ and the height of the CF change (Fig. 2) are underestimated by most models, it is likely that models do not simulate enough stratocumulus clouds. We also investigate how well GISS-E3, a model that can produce a decent spatial
pattern and amount of low clouds as well as a correct $\Delta CF$, perform against the observations. GISS-E3 $\Delta CF_{Cu}$ (green dotted line) is quite well captured by GISS-E3 whereas $\Delta CF_{Sc}$ (purple dotted line) is underestimated (its magnitude being out of the observed STD) and the Sc cloud top lifting is not reproduced. The overall $\Delta CF$ (orange dotted line) peaks slightly too high with a too small magnitude compared to the observations. Therefore, GISS-E3 likely underestimates the amount of Sc clouds compared to Cu, supporting the aforementioned hypothesis, in addition to slightly underestimating the amount of all types of
clouds in the present-day climate (Fig. 6a). The good agreement of the total $\Delta CREsw$ with the observations results from compensating errors between an overestimated Cu $\Delta CREsw$ and underestimated Sc $\Delta CREsw$ although the corresponding $\Delta LCCs$ are well simulated. This suggests that the regional GISS-E3 radiative effect of low-clouds is likely not well captured, in accordance with a long-standing problem in GCMs (e.g., Nam et al., 2012).

**5 Conclusion**

In response to interannual surface warming, the marine tropical low cloud cover (LCC) as observed by the active sensor from the CALIPSO satellite over a 10-year period significantly decreases ($\Delta LCC/\Delta SST = -3.59$ % K$^{-1}$). This reduction of the LCC is larger than that found using results from passive sensor satellites ($\Delta LCC = -1$ to $-2.95$ % K$^{-1}$), albeit consistent in terms of sign and magnitude (e.g., McCoy et al., 2017; Qu et al., 2015; Seethala et al., 2015). Overall, the ensemble mean of CMIP5 models captures the sign and the shape of the observed interannual low-cloud cover change ($\Delta LCC/\Delta SST$) quite well.
However, its magnitude is underestimated and the model variability is large ($\Delta LCC/\Delta SST = -2.25 \pm 1.58$ % K$^{-1}$), with some models (2 out of 14) even producing the wrong sign (a gain instead of a loss).

When scrutinized as a function of height, the interannual cloud fraction change ($\Delta CF/\Delta SST$) in the lower levels reveals various behaviors, which depend on the type of cloud and its height. We further show that it is possible to separate the model responses to SST variations using CALIPSO observations of the vertical cloud fraction ($\Delta CF/\Delta SST$) as a constraint: we select the GCMs
that produce the most realistic change in cloud profile per K of SST warming, referred to as "constrained" models. By doing so, we find that the "constrained" models, including the latest version of the GISS model (GISS-E3), simulate a more realistic





behavior of low-level cloud fraction and their associated interannual radiative feedbacks ($\Delta CRE_{SW}/\Delta SST$) together with a smaller variability in response to a surface warming. Their averaged $\Delta LCC/\Delta SST$ is within the observed uncertainty while they slightly underestimate the $\Delta CRE_{SW}/\Delta SST$. Meanwhile, the "unconstrained" category, which includes the CMIP5 version of the GISS model (GISS-E2), fails to reproduce the observed magnitude of both quantities by a factor of 3 to 4. The fact that

models that simulate moist processes within the PBL produce sustainable stratocumulus decks appears crucial to replicate the observed relationship between cloud, radiation and surface temperature.

Separating clouds between stratocumulus and trade cumulus categories helps us better quantify their contribution to global tropical low-level cloud change. The vertical structure of the change is indeed different in regions dominated by stratocumulus

clouds than in those dominated by cumulus clouds. Over the stratocumulus regions, the cloud top increases slightly, accompanied by a large decrease of the cloud fraction below, whereas over the trade cumulus regions, cloud fraction decreases to a smaller degree, but over its full vertical extent. As a result, the cloud cover change per unit SST change is smaller over trade cumulus regions than over stratocumulus regions ($\Delta LCC_{Cu}/\Delta SST$ = -3.62 % K$^{-1}$ compared to $\Delta LCC_{Sc}/\Delta SST$ = -5.32 % K$^{-1}$). Passive sensor observations confirm this result; although their overall $\Delta LCC/\Delta SST$ is consistently smaller regardless of

the SST dataset used (Fig. S4), mostly attributable to trade cumulus regions where passive sensors are less sensitive to broken cumulus. However, the derived slopes for trade cumulus and stratocumulus from active and passive methods are within the measurement uncertainty and cannot formally be distinguished (Fig. S4).

Finally, a region-based evaluation of the GISS-E3 model suggests that producing realistic global $\Delta CF$, $\Delta LCC$ and $\Delta CRE$ may

be the result of compensating errors between the Sc-dominated and Cu-dominated regions. However, it is difficult to determine with certainty whether the model is biased or not as we discriminate these cloud types by regions and not by actual type with the method used in this study. Future work will focus on developing a method to discriminate stratocumulus from trade cumulus clouds in satellite-based observations. By doing so, we will be able to assess the spatial distributions of these clouds and to evaluate the models more precisely. In addition, refining the contribution of additional cloud controlling factors may

advance our understanding of physical processes driving the change of cloud fraction in response to a warming climate.

**Data availability**

The GISS-E3 simulations can be made available upon request; the final version of GISS-E3 will be made part of the CMIP6 model archive. The CMIP5 simulations were downloaded from the World Data Center for Climate (WDCC) website

[http://cera-www.dkrz.de/]. CALIPSO-GOCCP observations were downloaded from the CFMIP-Obs website (http://climserv.ipsl.polytechnique.fr/cfmip-obs/Calipso_goccp.html). CERES-EBAF 4.0 TOA fluxes were downloaded on the CERES website (https://ceres.larc.nasa.gov/). The ERA interim dataset was downloaded from climserv





(http://climserv.ipsl.polytechnique.fr/). The NOAA OI v2, COBE-SST2 and ERSST v5 datasets were provided by the NOAA/OAR/ESRL PSD, Boulder, Colorado, USA, from their website (https://www.esrl.noaa.gov/psd/).

**Author contributions**

GC and AD designed the study and GC carried out the analysis. AD, AA, GE, MK, AF, YC and M-SY developed the model code and GC performed the simulations. GC wrote the manuscript with contributions from all co-authors.

**Competing interests**

The authors declare that they have no conflict of interest.

**Acknowledgements**

GC and AD were supported by a CloudSat-CALIPSO grant at the NASA Goddard Institute for Space Studies. AA, MK, YC, YM-S and AF were supported by a NASA Modeling, Analyis, and Prediction grants. The authors acknowledge the World Climate Research Programme's Working Group on Coupled Modeling, which is responsible for CMIP, and thank the climate modeling groups for producing and making available their model output

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

**Tables**

Table 1: **List of other models used in this analysis in addition to GISS-E2 and GISS-E3. Moist PBL means that the model simulates moist processes in the PBL, by either the turbulence, the convection or both parametrizations. The stars mark the constrained models as explained in section 4.1.**

| Model | Reference | PBL scheme | Moist PBL |
|---|---|---|---|
| BCC | Wu et al., 2014 | Holtslag and Boville, 1993 | no |
| CanAM4* | Von Salzen, 2013 | Revised Abdella and McFarlane 1996 | yes |
| CCSM4-CAM4 | Neale et al., 2010a | Holtslag and Boville, 1993 | no |
| CESM-CAM5* | Neale et al, 2010b | Bretherton and Park, 2009 | yes |
| CNRM | Voldoire et al., 2011 | Mellor and Yamada 1982 | no |
| GFDL* | Donner et al., 2011 | Lock et al., 2000, Anderson et al. 2004, Louis 1979 | yes |
| GISS-E2 | Schmidt et al., 2006 | Schmidt et al., 2006 | no |
| GISS-E3 Dev* | | Bretherton and Park, 2009 | yes |
| HadGEM2A* | Martin et al., 2011 | Lock et al., 2000, Lock et al., 2001, Brown et al., 2008 | yes |
| IPSL5A | Hourdin et al., 2006 | Louis 1879, Laval et al., 1981 | no |
| IPSL5B* | Hourdin et al., 2013 | Yamada et al., 1993, Rio and Hourdin, 2008, Rio et al., 2010 | yes |
| MIROC5 | Watanabe et al., 2010 | Revised Mellor and Yamada, 1982 | no |
| MPI | Stevens et al., 2013 | Brinkop and Roeckner, 1995 | yes |
| MRI | Yukimoto et al., 2012 | Mellor and Yamada, 1982 | no |



**Table 2: CRE, LCC and CRE/LCC changes depending on the cloud regime for the models and the observations in subsidence regimes defined as $\omega_{500} > 10$ hPa/day. The constrained models and the observations are represented in bold. The star means that the models include moist processes in the PBL (either due to turbulence parametrization, shallow convection or both). The numbers into parenthesis correspond to the standard deviation, computed based on four different SST datasets in the observations.**

|  | ΔLCC (% K-1) | | ΔCRE (W/m²/K) | | ΔCRE/ΔLCC (W/m²/%) | |
|---|---|---|---|---|---|---|
| BCC | -2.95 | | 1.91 | | -0.65 | |
| **CanAM4*** | **-4.51** | | **3.78** | | **-0.84** | |
| CCSM4 (CAM4) | -2.15 | | 0.29 | | -0.14 | |
| **CESM1-CAM5*** | **-2.88** | | **0.88** | | **-0.30** | |
| CNRM | 0.31 | | -2.51 | | -8.03 | |
| **GFDL*** | **-2.33** | | **2.24** | | **-0.96** | |
| GISS-E2 | 0.22 | | -1.77 | | -7.86 | |
| **GISS-E3*** | **-3.55** | | **2.94** | | **-0.83** | |
| **HadGEM2A*** | **-2.86** | | **1.98** | | **-0.69** | |
| IPSL5A | -0.73 | | 5.36 | | -7.39 | |
| **IPSL5B*** | **-4.90** | | **3.77** | | **-0.77** | |
| MIROC5 | -0.86 | | -1.00 | | 1.17 | |
| MPI* | -2.85 | | 2.89 | | -1.01 | |
| MRI | -1.59 | | 1.78 | | -1.12 | |
| | | | | | | |
| Multimodel Mean | -2.26 | 1.59 | 1.61 | 2.23 | -2.10 | 3.12 |
| Unconstrained | -1.32 | 1.28 | 0.87 | 2.63 | -3.13 | 3.90 |
| **Constrained** | **-3.51** | **1.01** | **2.60** | **1.13** | **-0.73** | **0.23** |
| | | | | | | |
| ***Obs*** | ***-3.59*** | ***0.28*** | ***3.05*** | ***0.28*** | ***-0.85*** | ***0.02*** |



**Table 3: CRE, LCC and CRE/LCC changes depending on the type of clouds for the different observational datasets used in the study. The numbers into parenthesis correspond to the standard deviation derived using four SST datasets.**

| Quantity | Obs | Type of Clouds | | |
| --- | --- | --- | --- | --- |
| | | all | Cu | Sc |
| $\Delta$LCC (% K-1) | GOCCP | -3.76 | -3.62 | -5.32 |
| | ISCCP | -2.95 | -2.31 | -5.22 |
| | ISCCP Q15 | -2.79 | -1.4 | -6.06 |
| | MODIS | -2.9 | -3 | -3.94 |
| | MODIS Mask | -1 | -0.34 | -2.05 |
| $\Delta$CRE (W/m$^2$/K) | CERES | 3.26 | 1.6 | 7.12 |
| $\Delta$CRE/$\Delta$LCC (W/m$^2$/%) | GOCCP | -0.87 | -0.44 | -1.34 |
| | ISCCP | -1.11 | -0.69 | -1.36 |
| | ISCCP Q15 | -1.17 | -1.14 | -1.17 |
| | MODIS | -1.12 | -0.53 | -1.81 |
| | MODIS Mask | -3.26 | -4.71 | -3.47 |





## Figures

**Figure 1: Geographic distribution of low cloud cover (LCC, %) for CALIPSO-GOCCP observations (a) and for GISS-E3 (b), GISS-E2 (d) and the multimodel, constrained and unconstrained models (f-h-j, respectively) along with their corresponding bias against CALIPSO-GOCCP observations (c-e-g-i-k, models minus CALIPSO-GOCCP). The blue contour denotes the regions wherein the**
5 **$\omega_{500}$ of each dataset (ERAI reanalysis for the observations, Dee et al., 2011) is greater than 10 h Pa/d.**



**Figure 2:** Vertical profiles of cloud fraction (a and c, CF in %) and interannual cloud fraction change due to SST variations (b and d, ΔCF/ΔSST in % K⁻¹) as observed by CALIPSO-GOCCP observations (orange line with circles), and as simulated by the 14 models. The green line in panels (a, b) and the blue and purple lines with triangles in panels (c, d) correspond to the multimodel mean of all, the constrained and the unconstrained models, respectively. The dotted line denotes the height (3.36 km) used to define the low-cloud cover in CALIPSO-GOCCP.



**Figure 3: (a) LCC change per K of SST warming (ΔLCC/ΔSST, % K⁻¹, y-axis) as a function of the present day LCC (%, x-axis) for the models and the CALIPSO-GOCCP observations (orange circle). The unconstrained and constrained models are represented in purple and blue squares, respectively, while the stars denote the two GISS model versions, GISS-E2 in the unconstrained category and GISS-E3 development in the constrained category. The triangles correspond to the multimodel mean of each category. The solid black line is the linear regression between LCC and ΔLCC for all models but the outlier. (b) same as (a) for four versions of GISS-E3 run along the GISS-E3 development with (black symbols) and without the simulator (red symbols). Note that while the present-day LCC is largely affected by the use of the simulator, the ΔLCC/ΔSST is not.**

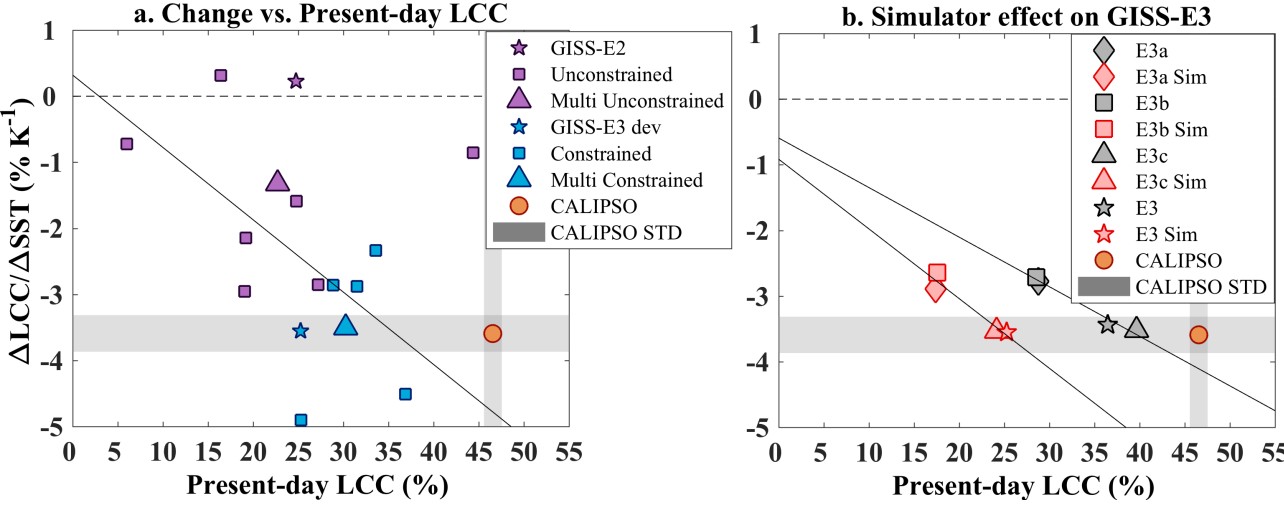





**Figure 4: Relationship between the ΔLCC/ΔSST (x-axis, % K⁻¹) and the ΔCRE/ΔSST (y-axis, W m⁻² K⁻¹) for the SW (a), the LW (b) and the net (c) radiation. The solid black line represents the linear regression of the models. The blue shading means a cloud cooling effect as opposed to red shading for a cloud warming effect.**

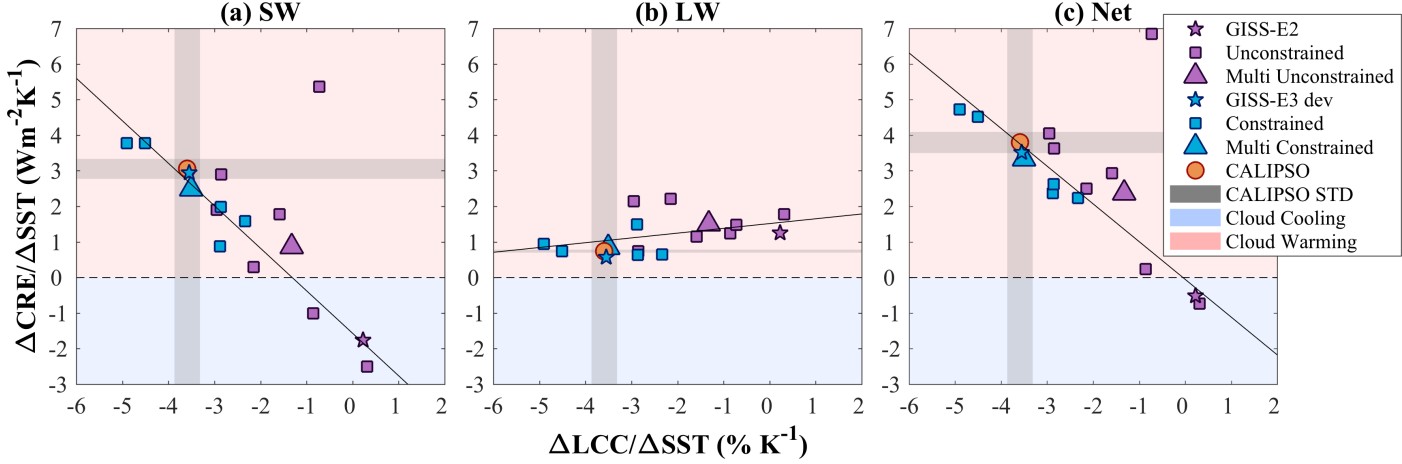



**Figure 5: (a) LCC change per K of SST warming (ΔLCC/ΔSST , % K⁻¹) per type of cloud (all clouds in orange, trade cumulus clouds in green and stratocumulus clouds in purple) for five observational datasets: CALIPSO-GOCCP (circles, 2007-2016), ISCCP (triangles, 1999-2008, Rossow and Schiffer, 1999), ISCCP Q15 (squares, 1999-2008, modified to take into account the shielding effect of high clouds following Qu et al., 2015), MODIS retrieval (stars, 2001-2015, Pincus et al., 2012) and MODIS mask (diamonds, 2001-**
5 **2015, using partly cloudy pixels, Pincus et al., 2012). Cloud types are defined based on the subsidence regime for "all" clouds (ω₅₀₀ > 10 hPa/d) and further based on four regions described in Fig. S3 for the trade cumulus (Cu) and stratocumulus clouds (Sc). The uncertainty bars correspond to +/- one standard deviation using the four SST datasets for the "all" type of cloud and to +/- one standard deviation of the four regions for the Cu and Sc types of cloud. (b) Same as (a) but with the EIS held constant as in Qu et al. (2015) and Klein et al. (2017).**

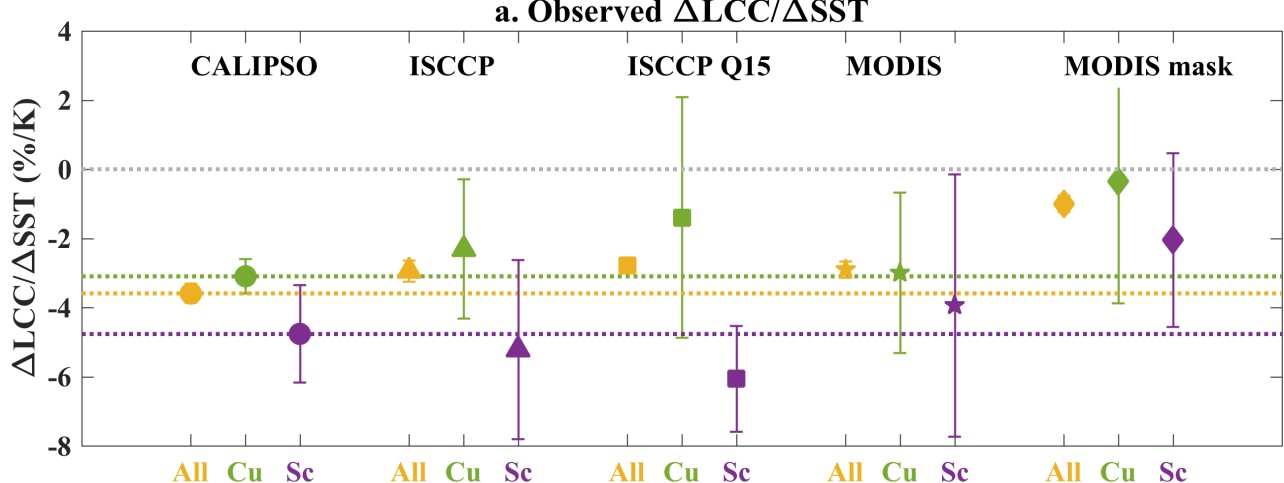

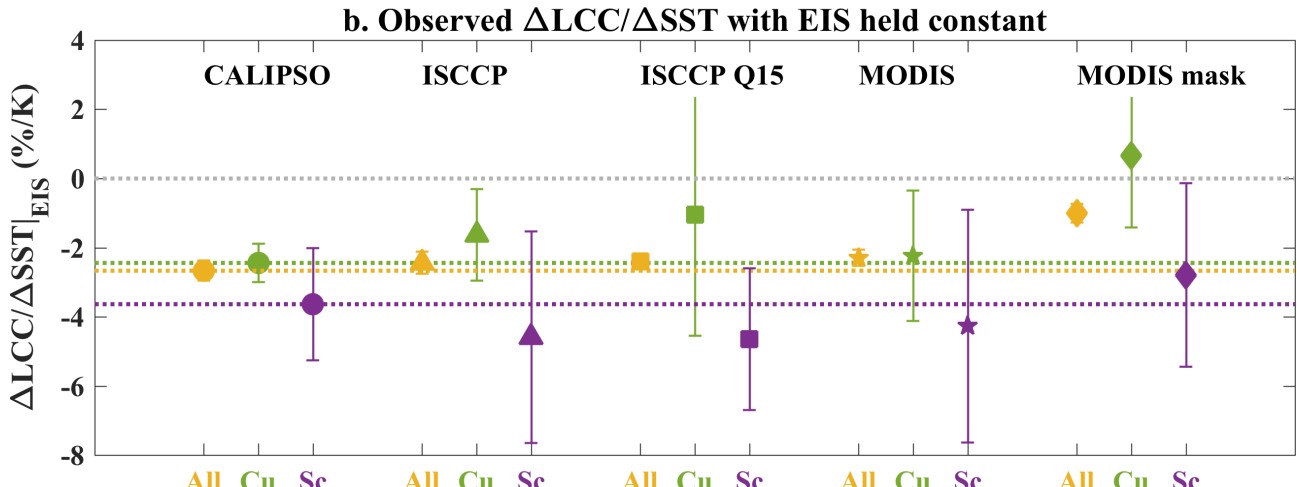



**Figure 6: (a) Vertical profiles (height in km, y-axis) of cloud fraction (CF in %, x-axis) and (b) interannual cloud fraction change due to SST variations (ΔCF/ΔSST in % $K^{-1}$, x-axis) as observed by CALIPSO-GOCCP observations for the three types of cloud: all clouds in orange, trade cumulus clouds (Cu) in green and stratocumulus clouds (Sc) in purple. The shading areas correspond to the standard deviation using the four SST datasets for the "all" type of cloud and to the standard deviation of the four regions for the Cu and Sc types of cloud. The horizontal dotted line denotes the height (3.36 km) used to define the low cloud cover in CALIPSO-GOCCP.**

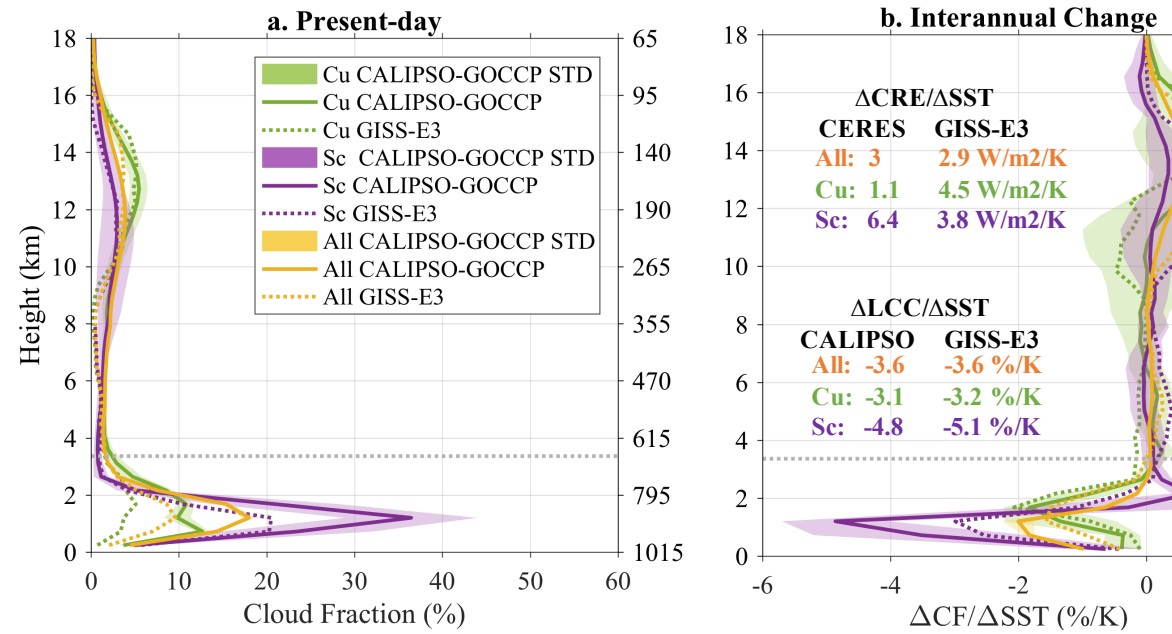