# Peer review of "Evaluating Models' Response Of Tropical Low Clouds to SST Forcings Using CALIPSO Observations"

_Atmospheric Chemistry and Physics, 2018_

## Referee Comment (RC1) · Anonymous Referee #1 · 13 Dec 2018

The authors compared observed low cloud cover (LCC) by active remote sensing instrument, CALIPSO lidar, and simulated LCC in CMIP5 models and two versions of GISS model with CALIPSO simulator implemented in those models. Then they classified the models into two groups based on vertical profiles of LCC and dLCC/dSST compared with the observations. The authors identified that the "constrained" model group tends to show larger decrease of LCC in response to SST warming than the remainders. The analysis, methods, and results are carefully described. The reviewer thinks that the results obtained here are worthwhile publishing as a research paper in this journal. However, I also have a major concern and a few minor comments. This paper's quality will be greatly improved if the authors take these comments into

account.

Major comment

In figures 3a and 4, inter-model relationships between present-day LCC and dLCC/dSST (or dLCC/dSST and dCRE/dSST) should not be simplified too much. The negative correlations are clearly found when MIROC5 (in the upper-right corners of these panels) is set aside. However, physical reasons why MIROC5 can be omitted from discussion here are not clearly explained. The authors should extend their notes here to physics-based discussion. Why is MIROC5 so unique among the 14 models? Is there any unique physics scheme implemented in that model? To obtain any physical explanations, the authors can contact the model developers and/or developers of CALIPSO simulator and discuss with them.

Specific comments

Page 2 line 2: You should combine the first and second paragraphs into a single paragraph

Page 9 line 14 "the radiative effect of increased CO2 on cloud-top turbulence": Any appropriate citations needed here. Any LES or GCM studies?

Page 13 line 6-12: Figure 7 of Su et al. (2013; doi:10.1029/2012JD018575) may be relevant to discussion here

Page 13 line 22, 33: "Finally" repeated

Figure 1 caption: "is greater than 10 h Pa/d" should be "equal to 10 h Pa/d over ocean". Please also check supplementary figure caption.

[Figure]

---

## Referee Comment (RC2) · Anonymous Referee #2 · 27 Dec 2018

Evaluating Models' Response Of Tropical Low Clouds to SST Forcings Using CALIPSO Observations

Cesana et. al.

Summary

This paper uses CALIPSO observations to evaluate examine the ability of global climate models (GCMs), mainly the GISS-E2 and GISS-E3 to simulate the observed marine low clouds in tropics and their response to changes in sea surface temperature (SSTs). The response is inferred using inter-annual variations in cloud and SST. It is found that model which better match the observations have a better match in their

response to SST variations. In addition to evaluating models, CALIPSO and passive observations are used to break down the response into those from different low cloud types, stratocumulus and cumulus.

The paper is generally well written and clear. I have only some minor comments below.

Specific comments:

Page 3, line 6: "2-dimensional swath" is a bit vague (x-y or x-z)?

Page 4, line 2: By using the nighttime only are you biasing the sampling to a particular part of the diurnal cycle? My understanding is that the GCMs sample all times of the day?

This reference examines the diurnal cycle of marine cloud feedback which might be of interest. It also examines the diurnal features of low marine clouds in some CMIP5 models.

Webb, M.J., Lock, A.P., Bodas-Salcedo, A. et al. Clim Dyn (2015) 44: 1419. https://doi.org/10.1007/s00382-014-2234-1

Page 4, line 23: Radiative balance during which period?

Section 2.2: Some time has elapsed since the manuscript was submitted. Is there a GISS-E3 paper available that you might be able to reference that contains most of this information?

Section 2.2: Is the required model output available for a longer period of time? It is mentioned a few times in the manuscript that a short time-period is used (a decade). If there is additional model output then is should be possible to indicate how well the 10 years period chosen represents a longer dataset (at least in the models).

Page 6, line 17: You might want to point out that the low-level cloud fraction is LCC referred to throughout the manuscript.

Page 8, line 32: "that is a too shallow PBL" -> "too shallow PBL"

Page 9, line 1: "strong masking effect". I thought the 500 hPa omega filtered out overlaying high clouds? In Figure S2 you show the cloud fraction profiles below 5 km. Could you not extend it vertically?

Page 9, line 9: But in the multimodel mean the response is similar to the observed?

Page 12, line 25: "2D" -> cloud-top properties?

Page 13, line 10: Do you get better results with EIS or other variants of LTS? You seem to use EIS later in the analysis (Figure 5).

---

## Author Comment (AC1) · 8 Jan 2019

Response to reviewer 1
The authors are very grateful for the helpful comments. Our responses are provided below in blue.

Major comment
In figures 3a and 4, inter-model relationships between present-day LCC and dLCC/dSST (or dLCC/dSST and dCRE/dSST) should not be simplified too much. The negative correlations are clearly found when MIROC5 (in the upper-right corners of these panels) is set aside. However, physical reasons why MIROC5 can be omitted from discussion here are not clearly explained. The authors should extend their notes here to physics-based discussion. Why is MIROC5 so unique among the 14 models? Is there any unique physics scheme implemented in that model? To obtain any physical explanations, the authors can contact the model developers and/or developers of CALIPSO simulator and discuss with them.
We agree that we did not provide a detailed explanation for the behavior of outliers in this section. Before addressing the reviewer concern, we want to emphasize that the outlier in Fig. 3a is MIROC5 whereas the outlier in Fig. 4a is IPSL5A. Although we think it is beyond the scope of our paper - focused in detail mainly on GISS model evaluation - to contact any modeling center to inquire about a particular behavior evident in our analysis, we do attempt to provide a physically-based explanation using the existing literature for each of these models to address the reviewer's concern.

First, regarding Fig. 3a MIROC5:
The slight correlation that exists between ΔLCC/ΔSST and LCC in Fig. 3a is tied to whether or not models simulate i) Sc clouds in the correct regions and 2) their transition to shallow cumulus clouds. First, to produce Sc clouds, a model needs a turbulent scheme that accounts for moist processes in the PBL in which clouds maintain the turbulent mixing that sustains them. From Watanabe et al. (2010), it appears that MIROC5 does not have a moisture-aware turbulence scheme. Therefore, the model does not produce large, persistent decks of low clouds in Sc regions (off the western coasts of continents) and its ΔLCC/ΔSST is small. Second, a model has to develop shallow convection over a warmer sea surface to vent the PBL and transition from Sc to shallow cumulus clouds. However, MIROC5 lacks any such transition owing to "insufficient vertical mixing of the humid air in the PBL and the dry air in the free troposphere (FT)" as pointed out by Ogura et al. (2017) and Tatebe et al. (2018). This insufficient mixing results in too much low-cloud cover in the trade-wind regions and small spatial variability over the tropical oceans (see Fig. R1), consistent with a relatively large LCC value in Fig. 3a, as well as a small amplitude of ΔLCC/ΔSST because SST changes poorly propagate through the PBL to the FT (consistent with what found by Sherwood et al., 2014).
*"The explanation of the MIROC5 behavior is twofold. First, similar to the other unconstrained models, MIROC5 evidently lacks the model physics to produce Sc-type clouds, i.e., simulating moist processes in the PBL in which stratiform clouds maintain the turbulent mixing that sustains them. As a result, its ΔLCC/ΔSST is small. Second, MIROC5 suffers from an insufficient vertical mixing of the humid air in the PBL and the dry air in the free troposphere (Ogura et al.,2017; Tatebe et al., 2018), which generates too large of an LCC over trade-wind regions compared to CALIPSO-GOCCP (not shown) and the largest mean LCC among the models. In addition, this poor vertical mixing may also explain the small amplitude of ΔLCC/ΔSST because SST changes*

*poorly propagate through the PBL to the free troposphere (consistent with what found by Sherwood et al., 2014)".*

[Figure]

*Figure R1: Maps of (a) low cloud cover (%) as simulated by MIROC5 through the lidar simulator and (b) the bias against CALIPSO-GOCCP. Note the lack of clouds over the Sc regions (attributable to the dry turbulence scheme) and the small spatial variability between the trade-wind regions and the rest of the tropics (attributable to insufficient vertical mixing between the PBL and the FT).*

Second, Fig. 4a, IPSL5A:

For the IPSL5A model, it is a different problem. Although the physics for Sc clouds is again lacking, the tuning strategy of IPSL5A was focused on getting close to the observed SW CRE at TOA (i.e., Hourdin et al., 2006: *"particular care was given to the tuning of the cloud radiative forcing, and in particular to its latitudinal variations"*). Because their cloud amount is so small, they had to make the clouds very bright to meet the observational target during their tuning process. This problem, also known as "the too few too bright" problem, explains the strong sensitivity of the $\Delta CRE_{SW}$ of IPSL5A model to $\Delta LCC$. To verify this, we computed 2D-histogram of SW CRE as a function of the respective LCC for all the models and for the observations in Figure R2. As expected, the IPSL5A model is the only model to show such a particular behavior, that is to say very low cloud amount having very large SW CRE. The deficiency is now better explained in the manuscript in section 4.3: *"This so-called "too few too bright" problem may explain the particular behavior of IPSL5A model in Fig. 4a. In this model, the SW CRE for a given LCC value is far too large compared to the observations and any other model (Fig. S6). This may be why the sensitivity of $\Delta CRE_{SW}$ to $\Delta LCC$ ($\Delta CRE_{SW}/\Delta LCC$, Tab. 2) is too large and far-off the correlation line in Fig. 4a."*

[Figure]

*Figure R2: 2D-histograms (frequency of occurrence, %) of shortwave cloud radiative effect (SW CRE, Wm⁻², y-axis) as a function of the low-cloud fraction (%, x-axis) for CERES-EBAF and CALIPSO-GOCCP observations (2007-2016) and for the 14 models. The bottom-right panel shows the averaged relationship (see the legend for models' name).*

In addition to the overestimation or underestimation of the CRE due to cloud property biases, we also note that the clear-sky portion of cloudy gridboxes may also influence $\Delta CRE/\Delta SST$. For example, artificially increasing the specific humidity of the clear-sky (for radiative transfer only) in GISS-E3 dampens $\Delta CRE_{SW}/\Delta SST$ compared to the control version of GISS-E3 because of the increased SW absorption by water vapor. As a result, the SW radiation reflected back to space is reduced, which reduces the SW CRE. Because the SW CRE is smaller, the ratio $\Delta CRE/\Delta LCC$ is reduced, meaning that the same reduction of LCC per K ($\Delta LCC/\Delta SST$) will generate a smaller SW positive feedback ($\Delta CRE_{SW}/\Delta SST$). We added this explanation in the manuscript: *"In addition, the radiative effect of clear-sky portion of the cloudy grid boxes can amplify or dampen the interannual SW cloud feedback ($\Delta CRE_{SW}/\Delta SST$). For example, artificially increasing the specific humidity of the clear-sky in GISS-E3 (for radiative transfer only) reduces the SW CRE at TOA because of the increased SW absorption by water vapor, which ultimately dampens the positive SW cloud feedback with respect to the change in LCC per K ($\Delta CRE_{SW}/\Delta LCC$, not shown)."*

Specific comments
Page 2 line 2: You should combine the first and second paragraphs into a single paragraph
Done.

Page 9 line 14 "the radiative effect of increased CO2 on cloud-top turbulence": Any appropriate citations needed here. Any LES or GCM studies?
This effect is documented by Bretherton 2015, which we added to the manuscript: *"For example, current climate variability does not include the radiative effect of increased CO2 on cloud-top turbulence, which may generate a reduction of stratocumulus cloud amount by increasing*

*downwelling LW flux and thus reducing cloud-top radiative cooling (e.g., Bretherton et al., 2015)."*

Page 13 line 6-12: Figure 7 of Su et al. (2013; doi:10.1029/2012JD018575) may be relevant to discussion here
We added the following sentence to the manuscript: *"This is somewhat consistent with the decoupling of the LTS and SST pointed out by Su et al. (2013)."*

Page 13 line 22, 33: "Finally" repeated
We removed it.

Figure 1 caption: "is greater than 10 h Pa/d" should be "equal to 10 h Pa/d over ocean".
Please also check supplementary figure caption.
We fixed this in the manuscript and the supplementary material.

References:
Ogura, T., Shiogama, H., Watanabe, M., Yoshimori, M., Yokohata, T., Annan, J. D., Hargreaves, J. C., Ushigami, N., Hirota, K., Someya, Y., Kamae, Y., Tatebe, H., and Kimoto, M.: Effectiveness and limitations of parameter tuning in reducing biases of top-of-atmosphere radiation and clouds in MIROC version 5, Geosci. Model Dev., 10, 4647-4664, https://doi.org/10.5194/gmd-10-4647-2017, 2017.

Sherwood, S. C., S. Bony and J.-L. Dufresne: Spread in model estimates of climate sensitivity traced to atmospheric convective mixing. Nature, 505, 37-42, doi:10.1038/nature12829, 2014.

Tatebe, H., Ogura, T., Nitta, T., Komuro, Y., Ogochi, K., Takemura, T., Sudo, K., Sekiguchi, M., Abe, M., Saito, F., Chikira, M., Watanabe, S., Mori, M., Hirota, N., Kawatani, Y., Mochizuki, T., Yoshimura, K., Takata, K., O'ishi, R., Yamazaki, D., Suzuki, T., Kurogi, M., Kataoka, T., Watanabe, M., and Kimoto, M.: Description and basic evaluation of simulated mean state, internal variability, and climate sensitivity in MIROC6, Geosci. Model Dev. Discuss., https://doi.org/10.5194/gmd-2018-155, in review, 2018.

---

## Author Comment (AC2) · 8 Jan 2019

Response to reviewer 2
We thank the reviewer for helping us improve our manuscript. Our responses are provided below in blue.

Specific comments:
Page 3, line 6: "2-dimensional swath" is a bit vague (x-y or x-z)?
The CALIPSO lidar laser produces a beam diameter of ~70 m at the surface every 333m along a polar orbit with an inclination of 98.2˚, which crosses the equator at approximately 0130 and1330 local time. The formulation used in the manuscript was confusing, so we changed the sentence to:
*" However, the narrow swath of the lidar – a beam diameter of 70 m every 333 m along-track – produces a much smaller sample of clouds than passive instruments. "*

Page 4, line 2: By using the nighttime only are you biasing the sampling to a particular part of the diurnal cycle? My understanding is that the GCMs sample all times of the day?
This reference examines the diurnal cycle of marine cloud feedback which might be of interest. It also examines the diurnal features of low marine clouds in some CMIP5 models. Webb, M.J., Lock, A.P., Bodas-Salcedo, A. et al. Clim Dyn (2015) 44: 1419. https://doi.org/10.1007/s00382-014-2234-1
It is true that the diurnal cycle affects LCC, which is maximum in the morning. However, in their supplementary material (text S1), Cesana and Waliser (2016) showed that the difference between all time and gridbox total-column cloud fraction average (cltcalipso) and the same quantity sampled along the CALIPSO orbit is negligible (less than 1%, absolute value) in a sample of four models. As a result, we consider it unlikely that using all times of the day rather than only 01h30 local time would significantly affect the $\Delta$LCC/$\Delta$SST relationship in the models and explain large differences between models and observations. In addition, Webb et al. (2015) found that temporal sub-sampling was not relevant to explaining the multi-model spread in cloud feedbacks. We now acknowledge this at the end of section 2.2: *Although the diurnal cycle of LCC is not fully represented in the observations (sampled at 0130 and 1330 local time), the total-column cloud fraction mean from the lidar simulator is not substantially different from that extracted along the CALIPSO footprint (<1% absolute difference; Cesana and Waliser, 2016) and effects on the strength of the cloud feedback have been found unimportant to understanding multi-model spread in overall cloud feedback (Webb et al., 2015).*
.

Page 4, line 23: Radiative balance during which period?
We added the missing information to the manuscript: *"Over 2007-2015, this version has a small positive radiative imbalance (0.29 W m$^{-2}$) of a few tenths of a W m$^{-2}$ less than that estimated for the real world in the early 21st Century (0.6 W m$^{-2}$).".*

Section 2.2: Some time has elapsed since the manuscript was submitted. Is there a GISS-E3 paper available that you might be able to reference that contains most of this information?
Unfortunately, no new paper is available as GISS-E3 is still under development, which is why we provide a somewhat detailed description here.

Section 2.2: Is the required model output available for a longer period of time? It is mentioned a few times in the manuscript that a short time-period is used (a decade). If there is additional model

output then is should be possible to indicate how well the 10 years period chosen represents a longer dataset (at least in the models).

We did not mention any of our sensitivity analysis on the time period chosen in this version of the manuscript. Because the COSP simulator is run offline in GISS-E3 it would require not insubstantial effort to analyze another 20 years. However, using the other models, we find (see Figure R1 below) that choosing a different time period (either the full AMIP period or the last 18 or 9 years) may slightly influence the ΔLCC/ΔSST by a few tenths of percent per K (in absolute value) in a subset of two unconstrained and two constrained models, which is far less than the model-to-observation difference, and makes almost no difference for the ΔCRE/ΔSST relationship. This result is now mentioned in the manuscript and in the supporting information (Fig. S1): *"Using a shorter or longer time period may affect the ΔLCC/ΔSST relationship by a few tenths of percent per K (absolute value, Fig. S1), yet it remains much smaller than the models' bias."*

[Figure]

*Figure R1: Relationship between ΔLCC/ΔSST (x-axis, % K-1) and ΔCRE/ΔSST (y-axis, W m-2 K-1) for SW radiation as in Fig. 4a of the manuscript. Here we study the sensitivity of that relationship to the chosen time period in four models. The results are shown for three periods of time: the full AMIP period (1979-2008, circles), the last 18 years as used in the manuscript (squares) and the last 9 years as used in GISS-E3 (triangles).*

Page 6, line 17: You might want to point out that the low-level cloud fraction is LCC referred to throughout the manuscript.
Done.

Page 8, line 32: "that is a too shallow PBL" -> "too shallow PBL"
Done.

Page 9, line 1: "strong masking effect". I thought the 500 hPa omega filtered out overlaying high clouds? In Figure S2 you show the cloud fraction profiles below 5 km. Could you not extend it vertically?
The reviewer is correct that we chose this omega500 threshold to reduce high-cloud– and it works perfectly in the observation as shown in Fig. S1. However, some models have a large high-cloud bias that may generate more masking effect than in the observations. We vertically extended Fig. S2 as requested by the reviewer to show this.

Page 9, line 9: But in the multimodel mean the response is similar to the observed?
This is correct, we modified the manuscript accordingly: *"As for the mean cloud profiles, ... while the multimodel mean captures the observed shape of ΔCF/ΔSST to some extent"*.

Page 12, line 25: "2D" -> cloud-top properties?
We acknowledge that this was ambiguous and changed it as suggested by the reviewer.

Page 13, line 10: Do you get better results with EIS or other variants of LTS? You seem to use EIS later in the analysis (Figure 5).
In GISS-E3, both the EIS and the LTS are well correlated with the LCC in the tropics. While EIS is available from GISS-E3, we could only compute the LTS with the output available from other models. We now mention this information in the manuscript: *"Using estimated inversion strength (EIS) – an LCC predictor that can be also used at mid-latitudes (Wood and Bretherton, 2006) – rather than LTS gives even better correlations in both the observations (see Fig. 5) and E3. However, we could only compute the LTS with the output available from other models."*